# Flexibility of Oxidized and Reduced States of the Chloroplast Regulatory Protein CP12 in Isolation and in Cell Extracts

**DOI:** 10.3390/biom11050701

**Published:** 2021-05-08

**Authors:** Helene Launay, Hui Shao, Olivier Bornet, Francois-Xavier Cantrelle, Regine Lebrun, Veronique Receveur-Brechot, Brigitte Gontero

**Affiliations:** 1Aix Marseille Univ, CNRS, BIP, UMR7281, F-13402 Marseille, France; lanqiqiaopan@163.com (H.S.); vbrechot@imm.cnrs.fr (V.R.-B.); 2NMR Platform, Institut de Microbiologie de la Méditerranée, Aix Marseille Univ, F-13009 Marseille, France; bornet@imm.cnrs.fr; 3CNRS, ERL9002, Integrative Structural Biology, Univ. Lille, F-59658 Lille, France; francois-xavier.cantrelle@univ-lille.fr; 4U1167, INSERM, CHU Lille, Institut Pasteur de Lille, F-59019 Lille, France; 5Plate-forme Protéomique, Marseille Protéomique (MaP), IMM FR 3479, 31 Chemin Joseph Aiguier, F-13009 Marseille, France; rlebrun@imm.cnrs.fr

**Keywords:** Calvin–Benson–Bassham cycle, conditionally disordered protein, intrinsically disordered protein, photosynthesis regulation

## Abstract

In the chloroplast, Calvin–Benson–Bassham enzymes are active in the reducing environment created in the light by electrons from the photosystems. In the dark, these enzymes are inhibited, mainly caused by oxidation of key regulatory cysteine residues. CP12 is a small protein that plays a role in this regulation with four cysteine residues that undergo a redox transition. Using amide-proton exchange with solvent, measured by nuclear magnetic resonance (NMR) and mass-spectrometry, we confirmed that reduced CP12 is intrinsically disordered. Using real-time NMR, we showed that the oxidation of the two disulfide bridges is simultaneous. In oxidized CP12, the C_23_–C_31_ pair is in a region that undergoes a conformational exchange in the NMR-intermediate timescale. The C_66_–C_75_ pair is in the C-terminus that folds into a stable helical turn. We confirmed that these structural states exist in a physiologically relevant environment: a cell extract from *Chlamydomonas reinhardtii*. Consistent with these structural equilibria, the reduction is slower for the C_66_–C_75_ pair than for the C_23_–C_31_ pair. The redox mid-potentials for the two cysteine pairs differ and are similar to those found for glyceraldehyde 3-phosphate dehydrogenase and phosphoribulokinase, consistent with the regulatory role of CP12.

## 1. Introduction

Redox regulation based on disulfide-dithiol exchanges constitutes a rapid and reversible post translational modification (PTM) that affects protein conformation. PTM in intrinsically disordered regions (IDRs) can confer flexibility, or in contrast order, thereby allowing rapid regulation of cellular processes. The proteins with disorder triggered by different stimuli, including redox, pH, and temperature, are called conditionally disordered proteins (CDPs) [1]. PTMs of CDPs can contribute to the diversification and functionality of proteomes, by regulating different properties of proteins, which is termed the “proteoform concept” [2]. Among these CDPs, some are particularly sensitive to redox changes, and these have been termed redox-dependent CDPs [3]. In these CDPs, key cysteine residues can form disulfide bridges under oxidizing conditions. These reversible thiol/disulfide bridge transitions imply that cysteine residues not only promote folding but are also key regulatory residues. Many examples of proteins containing IDRs, or of intrinsically disordered proteins (IDPs) that contain a redox-sensitive cysteine residue pair, have been described in the literature. To cite but a few examples, Hsp33, a redox-regulated chaperone, undergoes an order-to-disorder transition upon oxidation [4,5], and Cox17, a copper chaperone, and its partner Mia40, an intermembrane space mitochondrial protein, undergo a disorder-to-order transition upon oxidation [6,7].

The redox potential of chloroplasts becomes more reduced upon a dark to light transition and reciprocally more oxidized upon light to dark transitions. Many enzymes are regulated by different PTMs such as *S*-glutathionylation or *S*-nitrosylation, but mainly through redox modulators via thiol-disulfide interconversions involving the ubiquitous thioredoxins (Trx) [8,9,10]. Light has a dual role and activates enzymes involved in carbohydrate synthesis while inhibiting enzymes involved in their degradation, avoiding futile cycling. For instance, two enzymes that play a major role in the carbohydrate breakdown are inhibited in the light: glucose-6-phosphate dehydrogenase involved in the oxidative pentose phosphate pathway [11,12] and phosphofructokinase (PFK) involved in glycolysis [13,14]. For example, in *Arabidopsis thaliana*, the isoform PFK5 was shown to be reduced by Trx f and oxidized and activated by a recently identified NADPH-dependent Trx-like 2/2-Cys peroxiredoxin pathway, which is branched to the ferredoxin NADP reductase [14,15]. In contrast, the Calvin–Benson–Bassham (CBB) cycle that is responsible for CO_2_ assimilation and carbohydrate synthesis is inactive under dark and only operates in the light. Activation as well as inhibition of CBB enzymes have been shown to be very quick, within 30 s [16,17,18,19]. The mechanism of activation has been intensively studied and is mainly under the control of Trxs through the ferredoxin-Trx system [20]. The inhibition mechanism also involves the ferredoxin-Trx system as well as the newly described Trx-like 2/2-Cys peroxiredoxin pathway [15,20].

Redox regulation of enzymes of the CBB cycle can be either direct via their regulatory cysteine residues (e.g., phosphoribulokinase (PRK), sedoheptulose-1,7-bisphosphatase, fructose-1,6-bisphosphatase, [21]) or indirect via redox mediators such as the chloroplast protein CP12. This protein of about 8.5 kDa is able to switch this pathway on in the light or off in the dark [22,23]. CP12 has four cysteine residues that form two disulfide bridges in the dark. The formation of these intramolecular disulfide bridges is associated with a large increase in affinity of CP12 for two enzymes of the CBB cycle, and results in the formation of a ternary supramolecular complex. The *N*-terminal disulfide bridge C_23_–C_31_ is proximal with the region that is essential for PRK-association W_35_xxVEExxxxxxH_47_ [24], and the *C*-terminal disulfide bridge C_66_–C_75_ is within the region that interacts with glyceraldehyde-3-phosphate dehydrogenase (GAPDH) [25,26]. When embedded in this ternary complex, the two enzymes and the CBB cycle are inactive. In contrast, upon dark to light transition, the two disulfide bridges of CP12 become reduced and the ternary complex dissociates, releasing active enzymes allowing the CBB cycle to operate. CP12 is present in most photosynthetic organisms [27,28] and has features of a disordered protein. In the green alga, *Chlamydomonas reinhardtii*, as well as in other organisms, this protein is fully disordered when reduced (CP12_red_) and bears some secondary structural elements when oxidized, although it is still flexible (Figure 1) [29,30,31]. The *C*-terminal region that surrounds the C_66_–C_75_ disulfide bridge folds into a stable α-helical turn. It is preceded by a disordered linker and a highly dynamical *N*-terminal region that contains the C_23_–C_31_ cysteine pair. CP12 belongs to the CDP family and its conditional disorder allows for nuanced control of CBB enzymes and their fine regulation through binding processes. Besides the dark down regulation of CBB enzymes, this protein is a jack-of-all trades and can perform different functions in a photosynthetic cell [22,32].

Some in vitro structural data on CP12 have been obtained in the last few years (Figure 1) [25,26,29,31], but, as for many IDPs and CDPs, the flexibility of the protein in its physiological milieu remains to be confirmed [33,34]. Indeed, in cells, the molecular crowding and the presence of interacting partners can add a major contribution to the conformational sampling of an IDP, as it has been shown on well-studied IDPs such as tau and alpha-synuclein [35,36]. This is also the case for CDP, and the redox-dependent disorder-to-order transition of Mia40 and Cox17 has been shown in isolation and within HeLa cell cytoplasm [37,38]. Nevertheless, the effect of the complex physiological environment has never been studied on the model CDP, CP12.

The objective of this study was to monitor the structural and redox properties of both cysteine residues pairs of CP12 depending on the physical-chemical conditions. In order to understand better the redox-dependent conformational sampling of CP12 in vitro, the amide proton exchange rates were measured by Mass-Spectrometry (MS) and Nuclear Magnetic Resonance (NMR), as well as the temperature and pH variation of the NMR observables. The conformational states of both cysteine residues pair regions in CP12_ox_ and CP12_red_ were also observed under more physiological conditions, in *C. reinhardtii* cell extracts. We thus confirmed the distinct dynamical nature of both cysteine residues pair regions with all the possible interacting partners, PTM mediators, or solubilizing molecules present in the cell extracts. Since, as mentioned above, it is the CP12 redox transition that governs the GAPDH and PRK regulation, we monitored the kinetics of the redox transition of CP12 in isolation and within *C. reinhardtii* cell extracts and determined the oxidation-reduction midpoint potentials of its two pairs of cysteine residues.

## 2. Materials and Methods

### 2.1. Amide-Water Proton Exchange Kinetic Measurements by NMR

The kinetics of amide-water proton exchange was measured on CP12_red_ (in the presence of 20 mM DTT_red_) and CP12_ox_ in 50 mM sodium phosphate (NaPi) pH 6.5, 50 mM NaCl, 1 mM EDTA, 5% D_2_O, with traces of sodium trimethylsilylpropanesulfonate (DSS) by NMR on a 600 MHz Advance III spectrometer (Bruker, Ettlingen, Germany) equipped with a cryoprobe. We used the CLEANEX-PM [39] pulse sequence, with the following spin-lock delays (*τ_m_* in ms): 1.6, 3.2, 6.5, 9.7, 20, 39, and 81 and as a pseudo-3D experiment. The data were recorded at pH 7.0 and pH 6.0 and at 283 K. The signal intensities for each residue (*I_τm_*) was normalized against that in a fast-heteronuclear single quantum coherence (FHSQC) [39] spectrum (*I_0_*) acquired using the same parameters, that is, a ^15^N acquisition time of 42 ms, a ^1^H acquisition time of 243 ms, and 64 scans. The spectra were transformed in nmrPipe [40], and the cross-peak intensities measured using NMRFAM-SPARKY [41]. The increase of resonance intensities as a function of the spin-lock delay (*τ_m_*) was fitted to the following equation using a Levenberg–Marquardt nonlinear regression function in Octave [42]:(1)$$\begin{array}{c} \;\frac{I_{\tau m}}{I_{0}}=\frac{k}{R_{1A,app}\;+\;k\;-\;R_{1B,\;app}}\times \left\{\exp\left(-R_{1B,\;app}\times \tau_{m}\right)-\exp\left[-\left(R_{1A,\;app}+k\right)\cdot \tau_{m}\right]\right\} \end{array}$$
where *R_1B,app_* is the apparent water relaxation rate and was set to 0.6 s^−^^1^, and *k* (the amide-water exchange rate) and *R_1A,app_* (the amide relaxation rate) were optimized. The fits are shown in Appendix A. The exchange rate (*k_ex_*) was set to k/0.85 to take into account the water saturation effect in the CLEANEX-PM experiment [39,43]. The uncertainties on *k_ex_* values were obtained by repeating the optimization upon randomly varying the experimental integrals within 10% error (which is an overestimation of the experimental error). The intrinsic rates of exchange (*k_int_*) were predicted using SPHERE [44].

### 2.2. Gibbs Free Energy Derived from Protection Factors

The protection factor for CP12_ox_ (*P*) is calculated with:(2)$$\begin{array}{c} P=\frac{k_{ex,\;oxidized\;state}}{k_{ex,\;reduced\;state}} \end{array}$$

The change in Gibbs free energy (Δ*G*) for the backbone structure of oxidized CP12 is calculated from *P* with the following equation:(3)$$\begin{array}{c} \Delta G=-RT\ln\left(P\right) \end{array}$$
with *R* being the gas constant (1.987 × 10^−3^ kcal K^−1^ mol^−1^) and *T* being the temperature (283 K).

### 2.3. Amide–Water Proton Exchange Kinetic Measurements by MS

MS was also used to monitor the rate of exchange of proton amide with deuterated solvent. CP12_ox_ and CP12_red_ in 15 mM Tris pH 6.6, 50 mM NaCl were diluted 10× in either 10 mM potassium phosphate in 100% H_2_O, pH 7.0 for control “undeuterated” experiments, or 10 mM potassium phosphate in 99.96% D_2_O, pD 7.0 for the “deuterated” experiments. pH values of D_2_O solutions were adjusted to the corresponding pD values using the equation (pD = pHread + 0.40). The final concentration of CP12_ox_ and CP12_red_ was 17 μM and 20 μM, respectively. The samples were left from 10 s to 15 min for the exchange to occur on ice. At different times, the exchange reaction was quenched by diluting twice in pre-chilled 100 mM potassium phosphate, pH 2.66 containing 2 mM Tris(2-carboxyethyl)phosphine (TCEP), and immediately injected on a nanoACQUITY UPLC™ system with HDX technology (Waters Corporation, Milford, MA, USA). The proteins were digested online at 20 °C in an immobilized pepsin column (2.1 × 30 mm, Applied Biosystems, CA, USA) for 5 min in 0.1% formic acid/H_2_O at a flow rate of 100 µL min^−1^. Peptides were subsequently trapped and desalted online using an ACQUITY UPLC^®^ BEH C18 VanGuard™ Pre-column (1.7 µm, Waters Corporation, Milford, MA, USA) at 0 °C, then eluted into an ACQUITY UPLC^®^ BEH C18 column (1.7 µm, 1 mm × 100 mm, Waters Corporation, Milford, MA, USA) held at 0 °C, and separated with a linear acetonitrile gradient containing 0.1% formic acid. Mass spectra were acquired on a SYNAPT-G1 mass spectrometer (Waters, Manchester, UK) with an electrospray ionization source and lock-mass corrected by a Glu-fibrinogen peptide solution, in MSe mode, over the *m*/*z* range of 50–2000. “Undeuterated” peptides were identified using Protein Lynx Global Server software 3.1 (Waters, Manchester, UK). Deuterium uptake data for each peptic peptide from the “deuterated” experiments were automatically calculated using DynamX 2.0 software (Waters, Manchester, UK) and the results were manually checked.

### 2.4. Temperature Dependence of the NMR Chemical Shift and Signal Intensity

^1^H-^15^N FHSQC of CP12_ox_ (400 μM) in 50 mM NaPi pH 6.5, 50 mM NaCl, and with 20 mM DTT_ox_, 5% D_2_O were recorded at varying temperature (277 K, 284 K, 291 K, 298 K, 305 K, 313 K, 319 K), with a proton acquisition time of 243.3 ms and a ^15^N acquisition time of 42 ms on a 600 MHz and a 900 MHz Advance III spectrometer (Bruker, Germany) equipped with cryo-probes. When mentioned, one equivalent of copper (CuSO_4_) was added to the sample to catalyze oxidation, followed by the addition of 1 mM EDTA and dialysis. The samples were recorded in 20 mM Tris, and at varying pH values, the pH being increased by the addition of NaOH. The spectra were processed with nmrPipe [40] with Sine Bell window function. The spectra were referenced using the frequency of water (carrier frequency) at these temperatures [40], and controlled via the chemical shift of DSS, with uncertainty below 0.05 ppm. The chemical shift of all CP12 amide protons, as well as the signal intensity, were determined in NMRFAM-SPARKY [41] and exported in Octave [42]. The chemical shift dependence with temperature was fitted with the following equation: (4)δH 1 (T)=slope×T+c
where *c* is a constant, *T* is the temperature in *K*.

### 2.5. Thermodynamic of the Redox Transition

The redox mid-potentials for the transition for each pair of cysteine residues were probed by NMR as proposed in [45]. Aliquots of CP12 (50 µM) in 50 mM NaPi pH 6.5, 50 mM NaCl, 20 mM DTT pH 7 were prepared with varying ratios of *DTT_ox_* to *DTT_red_* in a glovebox under an anaerobic atmosphere. The samples were left overnight at ambient temperature to reach equilibrium. The ratios of *DTT_ox_* to *DTT_red_* were quantified by NMR using the proton signal intensity at 2.85 and 3.05 ppm (*DTT_ox_*) and at 2.6 ppm (*DTT_red_*). ^1^H-^15^N FHSQC of each sample was recorded. The electropotential for each sample was calculated from the ratio of the concentrations of *DTT_ox_* over *DTT_red_* using the Nernst equation [45]:(5)$$\begin{array}{c} E_{h,\;pH\;7}=E_{DTT,\;pH\;7}^{0}+\frac{RT}{nF}\times \ln\left(\frac{\left[DTT_{ox}\right]}{{\left[DTT_{red}\right]}^{2}}\right) \end{array}$$
where *E_DTT, pH7_* is the redox mid-potential of *DTT* (−332 mV [45,46]), *R* is the gas constant (1.987 × 10^−3^ kcal K^−1^ mol^−1^) and *T* is the temperature (283 K), n is the number of electrons (2), and *F* is the Faraday constant (95,484.6 J V^−1^ mol^−1^).

^1^H-^15^N FHSQC was recorded at 283 K for all samples, the data were transformed in nmrPipe [40], and the signal intensity were obtained in NMRFAM-SPARKY [41] at the frequency of reduced C_23_ (*δ*^15^*N*: 117.632 ppm, *δ*^1^*H*: 8.282 ppm), reduced C_31_ (*δ*^15^*N*: 119.451 ppm, *δ*^1^*H*: 8.205 ppm), reduced C_66_ (*δ*^15^*N*: 121.895 ppm, *δ*^1^*H*: 8.284 ppm), reduced C_75_ (*δ*^15^*N*: 120.418 ppm, *δ*^1^*H*: 8.471 ppm), oxidized C_66_ (*δ*^15^*N*: 115.784 ppm, *δ*^1^*H*: 8.696 ppm), and oxidized C_75_ (*δ*^15^*N*: 116.351 ppm, *δ*^1^*H*: 7.948 ppm). The signal intensity of the reduced resonance as a function of the electropotential was fitted in Octave using the following sigmoidal function [46]:(6)$$\begin{array}{c} \frac{I}{I_{0}}=\frac{1}{1+exp\left[d\times \left(E_{h,\;pH7\;}-E_{Cysteine}^{0}\right)\right]} \end{array}$$
where *d* is the steepness of the curve, which depends on the number of electrons (here 2) and the temperature (here ambient temperature). We have imposed *d* = 0.2 to fit our experimental data.

The signal intensity of the oxidized resonance as a function of the electropotential was fitted using the following sigmoidal function [46]:(7)$$\begin{array}{c} \frac{I}{I_{0}}=1-\frac{1}{1+exp\left[d\times \left(E_{h,\;pH7\;}-E_{Cysteine}^{0}\right)\right]} \end{array}$$

The same experiment was repeated at pH 8 in 30 mM Tris, 50 mM NaCl. The redox potential of *DTT* is expected to change as a function of pH using the following equation [45]: (8)$$\begin{array}{c} E_{DTT,\;pH\;8}^{0}=E_{DTT,\;pH\;7}^{0}-59.1\;mV \end{array}$$
and was modified consequently in Equations (5)–(7).

### 2.6. Monitoring the Kinetic of Oxidation

Twenty (20) mM *DTT_red_* was added to 1 mM CP12 (50 mM NaPi pH 6.5, 50 mM NaCl, pH 7, 5% D_2_O) and the sample was left overnight to ensure complete reduction of the disulfide bridges. The absence of the disulfide bridge was monitored by the acquisition of a ^1^H-^15^N FHSQC. The reducing agent was then removed by a PD10 column, followed by dialysis (using 10 kDa cut-off vivaspin concentrators) with a buffer to which air was bubbled for one hour to oxidize the buffer. A series of 20 min long ^1^H-^15^N FHSQC was then recorded at 293 K. All spectra were transformed in nmrPipe [40], and the signal intensity at a frequency of reduced C_23_, C_31_, C_66_, C_75_, and oxidized C_66_ and C_75_ were obtained using the autoFit.tcl script in nmrPipe [40]. The oxidized C_23_ and C_31_ resonances are broadened beyond detection [31]. The intensities were then transferred into Octave [42], and the decrease in intensity of the reduced resonances as a function of time were fitted with the following equation using the Levenberg–Marquardt nonlinear regression function: (9)II0=e−kred→oxt+c
where c is a constant and k is the rate constant of the oxidation of the cysteine residues pair.

The increases in intensity of the oxidized resonance as a function of time were fitted with the following equation:(10)II0=c−e−kred→oxt
where c is a constant and k is the rate constant of the oxidation of the cysteine residues pair.

### 2.7. Monitoring of the Kinetic of Reduction

Twenty (20) mM *DTT_red_* was added to 400 μM CP12_ox_ (50 mM NaPi pH 6.5, 50 mM NaCl, 1 mM EDTA, 5% D_2_O). A series of 20 min long ^1^H-^15^N FHSQC was then recorded at 293 K. The spectra were analyzed, and the data were fitted as above. The decrease of oxidized resonance intensity as a function of time was fitted with the following equation: (11)II0=e−kox→redt+c
where c is a constant and k is the rate constant of reduction of disulfide bridges.

The increases in intensity of the reduced resonance as a function of time were fitted with the following equation:(12)II0=c−e−kox→redt
where c is a constant and k is the rate constant of reduction of disulfide bridges.

### 2.8. Cell Extract Preparation

*C. reinhardtii* CC124 cells were cultured at ambient temperature (22 °C), with a 14 h light/10 h dark cycle, 110 rpm shaking for five days until they reached an absorbance at 680 nm of 1.24 (18.10^6^ cells mL^−1^). Fifty (50) mL of culture were centrifuged at 2000× *g* for 15 min at 4 °C and resuspended in 1 mL of buffer 30 mM Tris pH 7.0, 20 mM *DTT_red_*, protease inhibitor Complete EDTA free (Roche) following the recommended concentration. The sample was again centrifuged at 2000× *g* for 15 min at 4 °C and resuspended in 300 μL of the same buffer and D_2_O was added to the sample. The sample was sonicated twice for one min on ice. Fifteen (15) μL of CP12_red_ was added to reach a final concentration of 50 μM. The total protein concentration in the sample was measured using the Bradford assay [47]. ^1^H-^15^N FHSQC of the sample was recorded at 283 K, with proton acquisition time of 243 ms, and nitrogen acquisition time of 42 ms, on a 600 MHz Advance III NMR spectrometer equipped with a cryo-probe (Bruker, Germany). The data were transformed in nmrPipe [40] and signal intensity for each resonance was acquired in NMRFAM-SPARKY [41].

For the oxidized sample, the same procedure was applied with the following modification: *C. reinhardtii* cells were left for 24 h in the dark before collection. Twenty (20) mM DTT_ox_ was added to the sample instead of *DTT_red_*, as well as 0.1 mM 3-(3,4-dichlorophenyl)-1,1-dimethylurea (DCMU) (Durion). Fifteen (15) μL of CP12_ox_ was added to reach a final concentration of 50 μM. 

### 2.9. Diffusion Coefficient Determination

In the presence of cell extract, the translational diffusion coefficients of CP12 were recorded using DOSY-NMR selectively on the ^15^N labelled protein using the heteronuclear stimulated echo experiment (XSTE) from Ferrage et al. [48]. Ten experiments were recorded in a pseudo-2D fashion using bipolar square gradients of 1.4 ms (δ) of strength varying from 2 to 98% of the maximum gradient strength (G, G_100%_ = 0.5146 T m^−1^) with a fixed diffusion delay of 200 ms (Δ).

In the absence of cellular extract, the same experiment was used to probe for the translational diffusion delay of isolated CP12, and the result was compared with that obtained using standard bipolar stimulated echo experiment (STE) diffusion experiment using the same parameter. The translational diffusion coefficients were identical within 4% uncertainty.

The data were processed in nmrPipe [40] and fitted to the Stejskal–Tanner equation in Octave [42]:(13)$$\begin{array}{c} \frac{I}{I_{0}}=exp\left[D\times \left(\Delta -\frac{\delta }{3}\right)\times \left(\delta \cdot G\cdot \gamma_{H}\right)\right] \end{array}$$
where *δ*, Δ, and *G* are the gradient duration, diffusion delay, and gradient strengths defined above. *γ_H_* is the proton gyromagnetic ratio (2.67 × 10^8^ rad s^−1^ T^−1^). The hydrodynamic radius associated with the diffusion coefficient was determined using the Stokes–Einstein equation: (14)$$\begin{array}{c} r_{h}=\frac{k_{B}\cdot T}{6\pi \cdot D\cdot \eta_{\left(T\right)}} \end{array}$$
where *k_B_* is the Boltzmann constant, *T* is the temperature in Kelvin, and *η* is the viscosity.

### 2.10. Determination of ΔG of Binding

The Δ*G* for binding of CP12_ox_ to its partner was determined from the following equation:(15)$$\begin{array}{c} \Delta G=RT\times ln\left(\frac{\left[CP12\right]\left[partner\right]}{\left[complex\right]}\right)=RT\times \ln\left(K_{D}\right) \end{array}$$
where *R* is the gas constant (1.987 × 10^−3^ kcal K^−1^ mol^−1^) and *T* is the temperature (283 K). The dissociation constant for the CP12-PRK complex is 1.3 μM, for the CP12-GAPDH complex is 0.4 nM, and for the PRK-(CP12-GAPDH) complex is 60 nM [23].

## 3. Results

### 3.1. Structural Transition of the Region Encompassing the C_66_–C_75_ Disulfide Bridge upon Oxidation of the Isolated Protein

We first investigated the dynamics of CP12_red_ by measuring the amide proton exchange rate with H_2_O by NMR on CP12_red_ to probe for the protected and exposed backbone amide protons. The measured k_ex_ rates were related to very fast amide–water exchange (on the order of 6.10^−2^ to 3 s^−1^) and were only slightly lower than the predicted ones using SPHERE (Figure 2a, Appendix A), in a quasi-uniform manner. These data confirm that in its reduced state, CP12 is highly disordered [30] with amide proton exposed to the solvent, and the rate-limiting step of the exchange is the intrinsic proton exchange rate [49].

Upon oxidation, a smaller subset of protons was protected compared to the reduced state; they are all in the region that surrounds the C_66_–C_75_ disulfide bridge. These are protons from residues E_63_ to D_68_ and E_74_ to Y_78_ for which the exchange rates were beyond our detection limit, and A_69_ and A_72_ for which exchange rates were significantly lower than in the fully disordered CP12_red_. Assuming that the rate-limiting step for this exchange is the proton exchange (Appendix A), these exchange rates can be related to protection factors (Figure 2b). These protection factors are in good agreement with our previously described *C*-terminal structure [31], with the most protected amide being within the two small helical structures connected by the disulfide bridge (Figure 2c).

Interestingly, the variation of proton chemical shift according to the temperature for the same set of CP12_ox_ residues (E_63_, F_65_, K_67_, D_68_, A_69_, A_72_, C_75_, and R_76_) had a slope more positive than −4.6 × 10^−3^ ppm K^−1^ (Figure 2d, Appendix A), which is also indicative of a hydrogen bond [50]. Together, these data confirm that the *C*-terminal region surrounding the C_66_–C_75_ disulfide bridge folds in a helical turn stabilized by H-bonds upon oxidation [31].

### 3.2. Structural Transition of the Region Encompassing the C_23_–C_31_ Disulfide Bridge upon Oxidation of the Isolated Protein

In the oxidized state, most residues of the *N*-terminal region presented a kinetic of HN→H_2_O exchange similar to the disordered reduced state (Figure 2b, Appendix A), contrary to the residues of the stable *C*-terminal region. MS measurement of the proton-deuterium exchange on CP12_ox_ also confirmed that these *N*-terminal amide protons exchange very quickly with the deuterated solvent (Appendix A). The variation of chemical shift according to the temperature for these resonances all had more negative gradients than −4.6 × 10^−3^ ppm K^−1^ (Figure 2d), indicative of more labile protons [50], confirming the amide proton exchange measurements. These data indicated that the *N*-terminal region surrounding the C_23_–C_31_ disulfide bridge is highly unstable.

The temperature dependence of the proton chemical shift is expected to be linear for a proton in a stable conformational state [51]. For a high number of amide protons in CP12_ox_, the temperature dependence of the NMR frequency of amide protons of residues deviated from linearity, in particular those neighboring the dynamic *N*-terminal region; for example, the resonances assigned to residues A_5_, V_45_, as well as a high field-shifted glycine resonance that is putatively assigned to the single glycine of the *N*-terminal region G_26_ (Appendix A). This was also the case for several unassigned resonances that we have ascribed to the region L_8_-A_43_ [31]. Of note, a few residues of the *C*-terminal turn also had a non-linear temperature dependence of their amide proton chemical shift (L_62_, E_63_, A_69_, A_72_, R_76_, Appendix A).

The curvature in the temperature dependence of the chemical shift indicates a chemical exchange between two forms or more for which the relative population varies with temperature [51]. In line with these observations, two resonances were observed for the side chain indole amide of the single tryptophan residue at position 35 (Figure 3). One of these two resonances overlaid with that of CP12_red_ Trp-Hε (Figure 3a). The ^1^H chemical shift of this resonance had a linear dependence with a negative slope of −3.3 × 10^−3^ ppm K^−1^ (Figure 3c), which is close to the threshold for a disordered amide [50]. To ensure that this resonance, assigned to unfolded Trp-Hε, did not arise from a fraction of reduced proteins, Cu^2+^ was added to the sample to fully oxidize the protein and then removed before NMR acquisition. The spectrum remained identical (Figure 3b), indicating that the unfolded Trp-Hε resonance belongs to CP12_ox_. The chemical shift temperature dependence of the second resonance had a positive slope (+8 × 10^−3^ ppm K^−1^), indicative of folded conformation. Interestingly, the difference in frequency between these two resonances assigned to the same proton decreased at high temperature, and this relates to an increase in intensity (and decrease in linewidth) for the folded Trp-Hε resonance (Figure 3c, Appendix A). The difference in frequency also decreased at higher pH (Figure 3b), and the resulting resonances at pH 8 and 9 fell in a linear chemical shift pattern defined by the two above-mentioned unfolded and folded Trp-Hε resonances [52,53]. This behavior is strongly indicative of a chemical exchange between a disordered and a folded conformation. The broad linewidth of these two resonances, and their almost co-linear displacement, relates to an intermediate with a slow timescale for their interconversion, which can be complex (Figure 3f), and this interconversion becomes faster at higher temperature and higher pH [54].

### 3.3. Structural Properties of CP12_red_ in the Presence of C. reinhardtii Cell Extract

CP12 is located in a highly crowded organelle, the chloroplast, and we aimed at grasping the effect of the macromolecules present in its physiological environment on the structural states of CP12 in its reduced (light) and oxidized (dark) states. To mimic the chloroplast environment, we thus added crude cell extract at high concentration (15 to 19 mg mL^−1^ of protein) to purified ^15^N-CP12 such that the observed CP12 protein represents 2 to 3% of the total mass of protein. On the reduced and disordered protein, there was no significant chemical shift displacement, indicating that the protein remains in a disordered state (Figure 4b), with the exception of the last two resonances, E_79_ and D_80_. The resonances of the *N*-terminal residues up to K_17_ including the His-tag were broadened beyond detection compared to the isolated protein (Figure 4d). Other resonances were broadened in the presence of cell extract compared to the isolated protein: E_40_ and L_41_, K_48_ and K_49_, as well as few resonances of the *C*-terminal region (D_70_ to D_72_, C_75_, and the last two residues E_79_, D_80_). On the contrary, the region C_31_–A_34_ did not present any chemical shift variation, or specific line-broadening in the presence of cell extract. The translational diffusion coefficient at 4 °C of reduced ^15^N-CP12 in the presence of cell extract was less than 2 × 10^−11^ m^2^ s^−1^, which is significantly lower compared to that of the isolated reduced protein (9.3 ± 0.15 × 10^−11^ m^2^ s^−1^, Appendix A).

### 3.4. Structural Properties of the C_66_–C_75_ Disulfide Bridge in CP12_ox_ in the Presence of C. reinhardtii Cell Extract

The ^1^H-^15^N HSQC spectrum of oxidized and partially folded CP12 in the presence of cell extract also remarkably resembles that of the isolated protein (Figure 5b). As for CP12_red_, the His-tag and the *N*-terminal residues’ (up to D_14_) resonances were also broadened beyond detection in the presence of cell extract (Figure 5d). On the contrary, the *C*-terminal region (from K_53_ onwards) presented relatively narrow linewidths and identical chemical shifts as compared to the isolated protein. The resonances for the residues A_15_ to S_42_ were broadened beyond detection both in the isolated protein and in the presence of cell extract. On the isolated protein, resonance linewidths were large up to residue S_42_, but in the presence of cell extract, resonance linewidths were large for all residues until D_50_. The translational diffusion coefficient at 4 °C of oxidized ^15^N-CP12 in the presence of cell extract was 4 ± 0.5 × 10^−11^ m^2^ s^−1^, which is significantly lower compared to that of the isolated oxidized protein (9.8 ± 0.2 × 10^−11^ m^2^ s^−1^, Appendix A).

### 3.5. Thermodynamical Properties of the Redox Transition of Both Disulfide Bridges in Isolated CP12

CP12 is a CDP, and its function is related to the significant structural transition upon reduction or oxidation of the two cysteine residues pair. From the above results, we showed that the two disulfide bridges are located into two structurally distinguishable regions separated by a flexible linker. We monitored the thermodynamical properties of these two disulfide bridges using a titration with the redox mediator *DTT_ox_/DTT_red_* [45,55] on isolated CP12. Surprisingly, the use of the well-known redox mediator GSSG/GSH was not possible because the redox transition was not reversible with this mediator [45]. The standard redox mid-potentials at pH 7.0 were −284 mV for the C_23_–C_31_ cysteine pair (Figure 6a) and −291 mV for the C_66_–C_75_ pair (Figure 6b). The variation of the standard redox mid-potential of the *N*-terminal pair followed the expected pH-dependence for a disulfide bridge (−59.1 mV per pH unit [45], Figure 6a). On the contrary, the redox mid-potential measured at pH 8 for the *C*-terminal disulfide bridge was more negative than expected (Figure 6c), indicating a possible pH-induced stability for the *C*-terminal folded oxidized state.

### 3.6. Rate of Oxidation and Reduction of Both Disulfide Bridges in Isolated CP12

Using NMR spectroscopy, we followed the kinetic of oxidation or reduction of each disulfide bridge of isolated CP12 upon changing the redox potential of the solution by exchange into a buffer that has been bubbled with air or upon addition of 20 mM *DTT_red_*. The rates of oxidation for both disulfide bridges were identical, indicating their synchronization (Figure 6d, Appendix A). In contrast, the reduction of the *N*-terminal bridge was much faster than that of the *C*-terminal bridge, indicating asynchrony in their reduction (Figure 6e, Appendix A). These rates measured on the isolated protein are not representative of those in the cell, where redox transitions are catalyzed by the complex Trx network but reflect the structural stability of both regions encompassing the two respective disulfide bridges.

### 3.7. Reversible Redox Transition of CP12 in the Presence of C. reinhardtii Cell Extract

Upon addition of crude cell extract from cells exposed to 24 h of dark, the ^15^N-CP12_ox_ protein was highly unstable, and its ^1^H-^15^N spectrum was quickly converted and overlaid that of CP12_red_ within 20–40 min, even when 20 mM *DTT_ox_* was added. This indicates that the electron transfer of the photosynthesis was active in these crude cell extracts and can activate the Trx network, resulting in the reduction of DTT and CP12 when cells were transferred from dark to the spectrometer. We therefore added 0.1 mM DCMU, a PSII inhibitor, that inhibits photosystems II and electron transfer reactions, and this prevented CP12 reduction. 

We monitored the kinetic of the redox transition triggered by strong reducing or oxidizing agents (in the presence of DCMU not to interfere with the Trx network). When dark extracts were exposed to strong reducing conditions (20 mM Na_2_S_2_O_2_), the ^1^H-^15^N spectrum of the protein was quickly converted and overlaid that of CP12_red_ (Figure 4c). Reciprocally, when light extracts were exposed to oxidizing conditions (20 mM H_2_O_2_), the ^1^H-^15^N spectrum of the protein was quickly converted within 20 min and overlaid that of CP12_ox_ (Figure 5c). Together, these results show that both oxidized-to-reduced and reduced-to-oxidized transitions of CP12 are very fast in *C. reinhardtii* cell extracts, in contrast to isolated CP12, very likely resulting from the presence of Trxs in the extracts. 

## 4. Discussion

CP12 is present in the chloroplast of photosynthetic organisms from cyanobacteria, green and red algae, higher plants, as well as heterokonts [27,28,32]. Various functions have been assigned to this protein, in particular, the dark down-regulation of two CBB enzymes: GAPDH and PRK, as well as their fast re-activation upon dark-to-light transition [22]. Genetic and proteomic studies on higher plants and heterokonts suggest that CP12 has other functions, for example, related to stress regulation [32,56,57,58,59]. The enigmatic and multiple functions of CP12 is not an unusual property for a protein belonging to the CDP family [1]. In this study, the structural transitions of CP12 at the molecular level occurring upon change in redox potential, which is the basis for its proposed regulatory functions, are reported. 

### 4.1. Isolated CP12_red_ Is Intrinsically Disordered

In the light, photosystems I and II produce a strong reducing potential that affects the chloroplastic proteins via the Fdx-dependent Trx [60], and this network is probably the best-studied within the chloroplast [61]. Under reducing conditions, we recorded the amide proton exchange rate with solvent by NMR and MS for the purified CP12 and these data confirmed that the unfolded reduced state of CP12 is highly flexible (Figure 2a), confirming our earlier observations with SAXS and NMR spin-relaxation experiments [30,31]. The absence of pre-formed motif is also confirmed in this study by the homogeneous NMR resonance intensity for all CP12_red_ residues (Figure 4d). Indeed, the existence of a small proportion of pre-formed motifs would give rise to selective broadening of the NMR resonances [30,62]. Together, these data confirm that CP12_red_ is intrinsically disordered (Figure 7a).

### 4.2. The Two Disulfide Bridges Are in Regions with Distinct Structural Properties in Isolated CP12_ox_

In the dark, oxidizing conditions prevail in the chloroplast [61], principally because the source of electrons (water splitting) does not occur and so does not fuel the Fdx-dependent Trx system. Oxidation of chloroplast proteins is then catalyzed by both the Fdx-dependent and the NADPH-dependent Trx systems. In the case of CP12, the four cysteine residues form two disulfide bridges. Our results showed that the *C*-terminal disulfide bridge (C_66_–C_75_) is located in a region in which amide protons are highly protected against solvent exchange (Figure 2b). This confirms that this region, encompassing residues L_62_–Y_78_, is highly stable, as we have previously shown by NMR spin-relaxation measurements [31] (Figure 7b). The *N*-terminal disulfide bridge (C_23_–C_31_) is in a region that remains highly dynamic, as shown here by the absence of protection against amide proton exchange by NMR and MS (Figure 2b, Appendix A). As shown previously for *C. reinhardtii* and *A. thaliana* CP12 proteins, the residues of this long *N*-terminal region, encompassing residues L_8_–A_44_, gives rise to broad NMR resonances [29,31]. We used the resonances from the unique tryptophan side chain to probe for the origin of this line broadening. Two resonances are observed (Figure 3). One resonance that is the most downfield overlays with that of the same proton in CP12_red_. Its variation with temperature follows what is expected for a residue in a disordered region [50,51]. Additionally, this resonance is not observed at pH 9, and this is also indicative of fast amide proton exchange with water, and therefore an absence of structure [67]. The second resonance is always observed, even at high pH, and varies with temperature with a positive slope, which is indicative of hydrogen bonds (Figure 3) [50,51]. Together, these data confirm our previous SAXS analysis that suggested that the *N*-terminal region can be either disordered or folded [31]. When oxidizing conditions prevail in the chloroplast, the pH in the stroma decreases to pH 7 [68]. Such a drop in pH induced a displacement of the resonances arising from Trp-Hε within a linear chemical shift pattern defined by the above-mentioned two resonances with a larger difference in intensity (Figure 3b). This displacement is accompanied by a decrease in linewidth and an increase in signal intensity. This indicates that the speed of interconversion between the two states is modified at high pH and becomes slower in NMR intermediate exchange timescale (μs-ms) [52,53,69], as we have previously suggested based on NMR spin relaxation experiments [31].

Altogether, these data suggest that in its oxidized state, CP12 is in an atypical conformational ensemble, in which the *C*-terminal region (L_62_–Y_78_) is highly stable, whereas the *N*-terminal region (L_8_–A_44_) co-exists in folded and unfolded states in a complex exchange in the ms timescale (Figure 7b). In other word, the *C*-terminal region of CP12 undergoes a complete disorder-to-order transition upon increasing the redox potential, as described for several redox-dependent CDPs [3]. In contrast, the *N*-terminal region is not fully ordered in the oxidized state. For this region, the folded state is only partially stabilized under oxidizing conditions. Our previous SAXS analysis suggested a relative proportion for the disordered and ordered conformation of 40%:60%, and our current temperature- and pH-dependent analysis suggest that this equilibrium is strongly dependent on the physico-chemical conditions (Figure 7b). These two regions (folded *C*-terminal helical turn and highly dynamic *N*-terminal region) are separated by a flexible linker of ten residues (D_50_–D_60_) that also delimits two regions of interaction with different partners.

### 4.3. The Distinct Regions of CP12_ox_ Differ in Structural Dynamics and in Affinity for Their Interacting Partners

CP12_ox_ associates and inhibits PRK and GAPDH in a ternary complex [23]. The structure of this complex has been solved recently for proteins from *A. thaliana* [26] and from the cyanobacterium *Thermosynechococcus elongatus* [25]. In these structures, the *C*-terminal region of CP12_ox_ folds into a helical turn that is identical to the stable structure that we have computed from NMR data [31]. The very high affinity for the *C*-terminal region of CP12 for GAPDH (sub nM) relates to a low Δ*G* of −12.7 kcal mol^−1^, a value that is below the mean value for complexes between two ordered proteins from ~200 studies [70] (Figure 7c). In contrast, the *N*-terminal region has a much lower affinity for PRK (−1.3 μM) [23], which relates to a higher ΔG of −8 kcal mol^−1^. This value of ΔG is identical to the mean value for complexes between an ordered and a disordered protein [70]. The CP12_ox_ binding region with PRK is located in the unstable and highly dynamic *N*-terminal region [24]. In line with the above-described structural dynamic for this region in the isolated protein, in the crystal structure of the PRK-CP12-GAPDH from *A. thaliana* [26] and in the cryo-electron microscopy structure from *T. elongatus* [25], the least resolved regions are the interaction surfaces between CP12 and PRK. Altogether, the interface between Nter-CP12_ox_ and PRK seems to be highly dynamic contrary to the Cter-CP12_ox_ interface with GAPDH, and these result from the relative structural stability of these two regions.

The two structurally and functionally distinguishable regions of CP12, a stable *C*-terminal helical turn and an unstable *N*-terminal region, fused via a flexible linker, offer a complex and tightly controlled mechanism to regulate the two binding partners. Interestingly, photosynthetic organisms possess CP12 homologs where only the *C*-terminal region (C_66_xxxP_70_xxxC_75_) is fused to other enzymes such as GAPDH-B isoforms [71] or adenylate kinase, ADK3 [72,73]. This latter protein also binds to GAPDH with a high affinity. In addition, other organisms possess CP12 homologs that only contain the conserved *N*-terminal motif A_34_WxxVEEL_41_ that is embedded in the dynamical PRK-binding region defined above [27,32], but the function of these proteins remains to be elucidated.

### 4.4. Thermodynamically Independent and Reversible Redox Transition of Both Disulfide Bridges in CP12

As mentioned above, the principal known function for CP12 is the redox regulation of CBB enzymes in the chloroplast. Here, we report on redox titrations by NMR that demonstrated that the four conserved cysteine residues of CP12 from *C. reinhardtii* could form two intermolecular disulfide bridges with different redox mid-potentials (Figure 7d). The redox mid-potential for the C_23_–C_31_ cysteine residues pair ($E_{pH\;8\;C.r\;N-\mathrm{ter}}^{0}$ = −343 mV) is more negative than those measured for the *N*-terminal disulfide bridge of *A. thaliana* CP12 isoforms ($E_{pH\;7.9\;A.t\;N-\mathrm{ter}}^{0}$ = −326 mV in the two isoforms CP12-1 and CP12-2 and $E_{pH\;7.9\;A.t\;N-\mathrm{ter}}^{0}$ = −332 mV in the isoform CP12-3 [65]), analyzed from redox titrations using 5,5′-dithiobis(2-nitrobenzoic acid) to detect thiol groups. The redox mid-potential for the *C. reinhardtii C*-terminal disulfide ($E_{pH\;8\;C.r\;C-\mathrm{ter}}^{0}=$ −356 mV) is intermediate with that found for the isoforms CP12-1 and CP12-2 of *A. thaliana* ($E_{pH\;7.9\;A.t\;C-\mathrm{ter}}^{0}$ = −350 mV) and that found for the isoform CP12-3 ($E_{pH\;7.9\;A.t\;C-\mathrm{ter}}^{0}$ = −373 mV) [65]. These more negative values in the algal CP12 redox mid-points contradict the previous observation that the redox potentials of Trxs in the alga are less negative than the higher plant counterpart [74]. Whether or not the different techniques are responsible for these differences needs to be investigated. Above all, and similar to what has been observed for *A. thaliana*, our results indicate that the *N*-terminal disulfide in *C. reinhardtii* requires fewer reducing conditions to dissociate than the *C*-terminal disulfide, i.e., in thermodynamic terms, the *N*-terminal disulfide is easier to reduce than the C terminal disulfide.

During light to dark transition, partially oxidizing conditions would cause the formation of the *C*-terminal disulfide of CP12 ($E_{pH\;8}^{0}$ = −358 mV) and then favor the formation of the binary complex A_4_-GAPDH-CP12 (Figure 7c,d). At these electropotential values, the A_4_-GAPDH tetramer from *A. thaliana* ($E_{pH\;7.9\;A4-\mathrm{GAPDH}}^{0}$ = −335 mV) would be reduced, but the A_2_B_2_-GAPDH tetramer that contains a CP12 homolog extension would be oxidized ($E_{pH\;7.9\;A2B2-\mathrm{GAPDH}}^{0}$ = −359 mV) [64]. The redox mid-potential of CP12 *C*-terminal bridge is consistent with the hypothesis proposed in [17] that CP12 is a redox mediator for GAPDH regulation. On transition to darkness, oxidation of Trxs would lead to the formation of the *N*-terminal disulfide bridge in CP12 ($E_{pH\;8\;N-\mathrm{ter}}^{0}$ = −343 mV), allowing the PRK-CP12-GAPDH to form, and afterwards, the regulatory disulfide in PRK would form ($E_{pH\;7.9\;\mathrm{PRK}}^{0}$ = −333 mV) [63]. These results provide key insights into the biological assembly and regulation of this ternary complex.

### 4.5. Asynchrony of the Reduction of the Two Disulfide Bridges, Synchrony of Their Formation

From a kinetic point of view, the rates of oxidation of the cysteine residues at the *N*- and *C*-terminal extremities were similar, which validates the above discussion on equilibrium. In contrast, the rates of reduction were different (Figure 7e). The synchrony of oxidation can be explained by the fully disordered nature of the reduced form. Similarly, the asynchrony of reduction can be explained by the dual nature of CP12_ox_ with a very dynamic *N*-terminal end, being prone to fast reduction, and a stable *C*-terminal helical turn less prone to reduction. On the isolated CP12, these processes, oxidation and reduction, were slow. In contrast, in cell extracts, that is, in CP12 physiological environment with active cellular redox-mediators, the redox transitions were much faster [17]. It would be interesting to compare the rate of thiol reduction in the cell (kinetics of CP12 thiol to disulfide transition) with the rate of reduction of the electropotential in the chloroplast upon dark to light transition [15], as this could provide a further fine regulation of CBB enzyme activity.

### 4.6. Effect of Cell Extract on N_ter_-CP12 and C_ter_-CP12

Within the chloroplast, redox modulation is clearly open to modification by a number of other parameters such as stromal pH, Mg^2+^, and metabolite concentrations, and above all, crowding. The chloroplast contains numerous phases including starch granules, thylakoid membranes [75], pyrenoid (the liquid non-membranous compartment containing the ribulose-1,5-bisphosphate carboxylase oxygenase) [76,77], as well as natural deep eutectic molecules with an extremely high local concentration [78,79,80]. The physico-chemical conditions prevailing in the chloroplast stroma therefore have to be taken into account when trying to unravel the processes of protein folding. CP12 is present in the chloroplast of *A. thaliana* both in the light and the dark [81]. We attempted to characterize the effect of the chloroplast environment on both CP12_red_ and CP12_ox_ by adding *C. reinhardtii* cell extract to purified ^15^N-CP12. The cytoplasm of *C. reinhardtii* is very limited in volume compared to other eukaryotic cells [82], so chloroplast molecules contribute to a major fraction of the cell extract. In addition, the interaction network of CP12 is expected to be present in cell extract as it has been previously shown that sonication does not break the networks of interactions between partners [83]. 

We observed the structural state of CP12 in a large excess of *C. reinhardtii* cell extract (2 to 3% of protein are ^15^N-CP12 in a total protein concentration of 15 to 19 mg mL^−1^). We observed that CP12_red_ in cell extract remains disordered, with a few residues being affected. Some residues present specific line-broadening of their NMR resonances (for example the *N*-terminal residues, E_40_–L_41_, K_48_–K_49_), and this could be ascribed to specific binding of molecules with CP12_red_. In the literature, no binding partners for CP12_red_ have been identified, possibly because, if they exist, their affinity to CP12 is weak, as is often the case for disordered proteins [70]. These affinities might increase within the cell, giving rise to the observed line-broadening. Indeed, molecular crowding is known to increase the equilibrium constants of association events [84,85]. For example, the association equilibrium constant for the dimerization of a 40 kDa monomer is 8 to 40-fold higher if the protein is in *E. coli* cytoplasm, where the concentration of proteins is extremely high (200 to 320 mg mL^−1^), compared to the same phenomenon in an aqueous solution [86].

In the presence of oxidized cell extract, only the stable *C*-terminal helical turn is observed in the NMR spectrum (Figure 5). This could be because the region concerned with line-broadening ascribed to the millisecond-timescale equilibrium on the isolated protein (L_9_–A_43_) is expanded in the presence of cell extract (*N*-terminal to D_50_). CP12_ox_ thus behaves like many proteins that were shown to be more thermodynamically stable in the presence of crowding agents because of excluded-volume effects and higher chances of intra-molecular interactions [87]. Another origin for line broadening could be the association within a macro-molecular complex as discussed above for CP12_red_. 

The translational diffusions of both CP12_ox_ and CP12_red_ are also dampened by the presence of *C. reinhardtii* cell extract (Appendix A) and are reduced two-fold for CP12_ox_ and by more than five-fold for CP12_red_. The macromolecular concentration in our experiment was 15 to 19 mg mL^−1^ of protein, and an increase in viscosity is thus expected. Ordered proteins are expected to be more affected by the increase in viscosity than disordered proteins [88,89], but this is not the case for the partially folded CP12_ox_ that is less affected than the disordered CP12_red_. A succession of association-dissociation events with several chloroplast partners can also explain this very slow diffusion. The presence of broadened resonances in both spectra, as discussed above, indicates the presence of such association-dissociation events. Surprisingly, diffusion of CP12_red_ is more affected than CP12_ox_, and this may suggest a higher number of weak (specific or not) interaction events for the disordered reduced state.

## 5. Conclusions

We have shown here that CP12 can regulate association/dissociation with two of its known partners via a range of equilibria: a folding equilibrium that is modulated according to redox potential, pH and temperature, as well as two binding equilibria. Because these regulatory processes respond to subtle changes in the physico-chemical conditions of the CP12 environment, it is important to report on these in more physiological relevant conditions, as we have initiated here using cell extracts. Performing in cell NMR would be ideal, as this has been done for bacteria, yeast, oocyte, and mammalian cells [90], but it has remained a real challenge for photosynthetic cells [91]. The main question remains: where is CP12 in the chloroplast? Protein spatial organization in *C. reinhardtii* chloroplasts have been described by Mackinder et al. [75,92] but these studies have not localized the enigmatic CP12 protein. Moreover, to complete our preliminary attempt to reconstitute the chloroplast environment, further investigations are required to understand the structural mechanisms that promote highly regulated unfolding-folding transitions under specific physiological conditions, including not only redox but also pH, temperature, and metabolites to finely tune the photosynthetic metabolism.

## Figures and Tables

**Figure 1 biomolecules-11-00701-f001:**
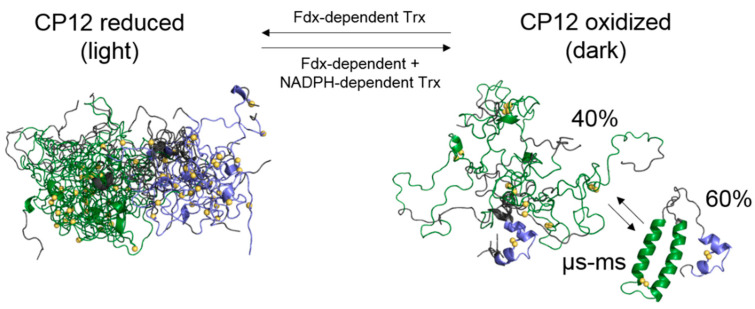
Model of the redox-transition of CP12. The structural models are derived from Launay et al. [30,31]. The complex Trx network is derived from Cejudo et al. [15]. The sulphur atoms of the two pairs of cysteine residues are in yellow. Trx and Fdx stand for thioredoxins and ferredoxin, respectively.

**Figure 2 biomolecules-11-00701-f002:**
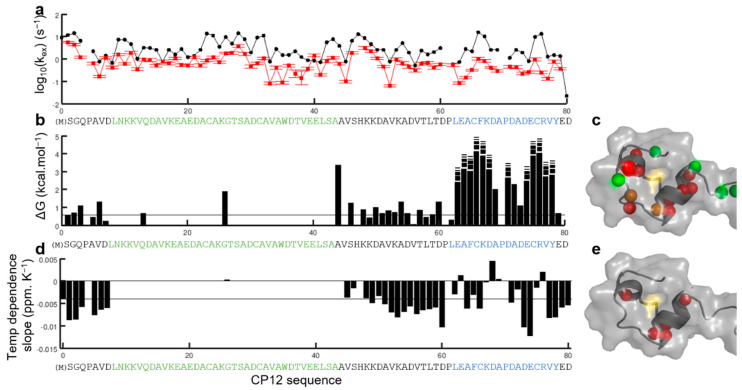
Labile amide protons in CP12_red_ and CP12_ox_. (**a**) Solvent-amine intrinsic proton exchange rates calculated using SPHERE [44] (black) and solvent–amide proton exchange rates measured using CLEANEX-PM experiment [39,43] on CP12_red_ (red). The similarity between both values indicated that CP12_red_ is intrinsically disordered. The signal intensity as a function of the spin-lock delay in the CLEANEX-PM experiments is shown in Appendix A. (**b**) Gibbs free energy of CP12_ox_ backbone amides derived from the measured solvent–amine proton exchange using CLEANEX-PM on CP12_ox_ versus CP12_red_ (refer to Material and Methods for details), assuming that the exchange falls into the EX2 regime that is highly probable at this low pH (7). The measured rates of exchange for CP12_ox_ amide at pH 7 and pH 6 are shown Appendix A. In (**b**,**d**), the stars indicate residues for which the resonances are broadened beyond detection in CP12_ox_ [31], and the disulfide bridges are indicated above the sequence. (**d**) Value of the slope of the temperature dependence of the proton chemical shift for CP12_ox_ residues. The line at −4.6 ppm K^−1^ indicates the threshold below which the data indicate the absence of a hydrogen bond [50]. The temperature dependence curves of the chemical shift of the amine proton for all CP12_ox_ residues are shown Appendix A. (**c**) Structure of the stable *C*-terminal helical turn with the amide protons protected from the exchange with water highlighted in red, and those exposed to exchange in green. (**e**) On the same structure, the amide protons for which the temperature dependence of the chemical shift has a positive slope are highlighted in red. The CP12 sequence is indicated below the graphs, with the following color coding: the residues that fold in the stable helical turn in the *C*-terminal region of CP12_ox_ are in blue, and the residues that undergo chemical exchange in CP12_ox_ are in green [31].

**Figure 3 biomolecules-11-00701-f003:**
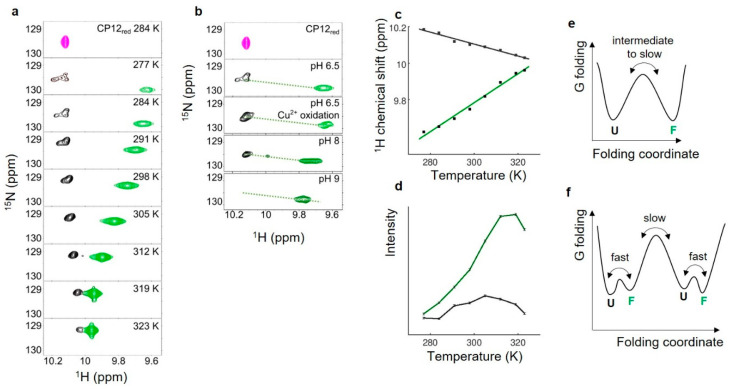
Temperature dependence of CP12_ox_ Trp-ε resonances indicating a chemical exchange. (**a**) Series of ^1^H-^15^N spectra of CP12_ox_ recorded at varying temperatures (277 K, 284 K, 291 K, 298 K, 305 K, 313 K, 319 K) and at pH 6.5 and plotted with the same contour levels. To ease the reading, the contour levels of each resonance have different colors. In (**a**,**b**), above the series, the spectrum of CP12_red_ at 284 K is shown (magenta). (**b**) ^1^H-^15^N spectra of CP12 recorded at 284 K at varying pH (pH 6.5 as in A, pH 8, and pH 9). An additional spectrum is shown at pH 6.5 where the protein has been pre-treated with Cu^2+^ to ensure full oxidation. The paramagnetic ion was removed by dialysis before NMR acquisition. (**c**) Temperature dependence of the proton chemical shift for each of these CP12_ox_ Trp-ε resonances using the same color coding (Appendix A). (**d**) Temperature dependence of the resonance intensity for each of these CP12_ox_ Trp-ε resonances using the same color coding (refer to Appendix A). (**e**,**f**) Gibbs free energy of folding for the Trp-ε residue, with two possible interconversion models. The timescales of interconversion are indicated above the arrow.

**Figure 4 biomolecules-11-00701-f004:**
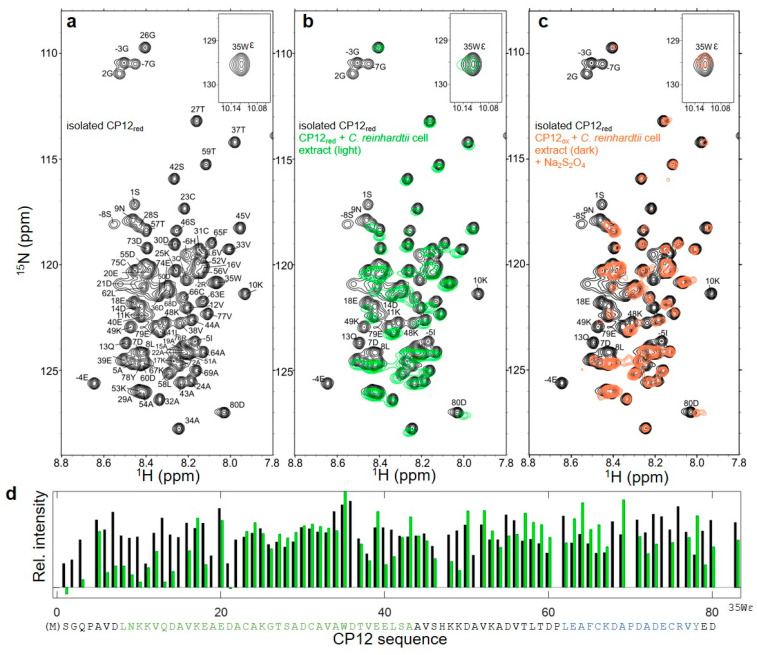
CP12_red_ in isolation and in the presence of *C. reinhardtii* cell extract. (**a**) ^1^H-^15^N HSQC spectrum of isolated CP12_red_. The region of the tryptophan side chain is shown in the insert. (**b**) Overlay of the ^1^H-^15^N HSQC spectra of isolated CP12_red_ (black) and that of CP12 in the presence of *C. reinhardtii* cell extract (green, molecular crowding corresponding to 15 mg mL^−1^ of protein). The data are recorded in the presence of 20 mM DTT_red_. (**c**) ^1^H-^15^N HSQC of CP12 in the presence of *C. reinhardtii* cell extract collected in the dark (molecular crowding corresponding to 19 mg mL^−1^ of protein), 0.1 mM DCMU, and 20 mM dithionite as a strong reducing agent. (**d**) Signal intensities for all residues in the ^1^H-^15^N HSQC spectrum of isolated CP12_red_ (black), and in the ^1^H-^15^N HSQC spectrum of CP12 _red_ in the presence of *C. reinhardtii* cell extract (green). Signal intensities were normalized against the mean of all intensities. The CP12 sequence is indicated below the graph with the same color coding as in Figure 2.

**Figure 5 biomolecules-11-00701-f005:**
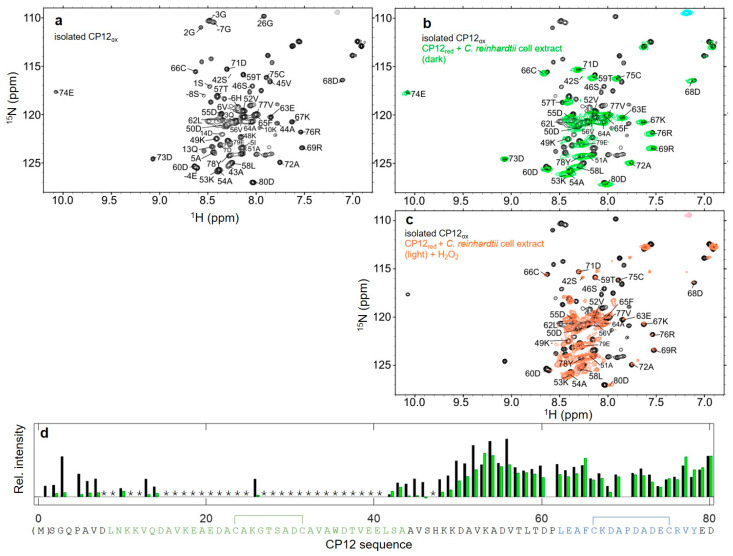
CP12_ox_ in isolation and in the presence of *C. reinhardtii* cell extract. (**a**) ^1^H-^15^N HSQC spectrum of isolated CP12_ox_. (**b**) Overlay of the ^1^H-^15^N HSQC spectra of isolated CP12_ox_ (black) and that of CP12 in the presence of *C. reinhardtii* cell extract (green, molecular crowding corresponding to 19 mg mL^−1^ of protein). The data is recorded in the presence of 20 mM DTT_ox_, and 0.1 mM DCMU. The assignments of the resonances that are observed in the presence of cell extract are indicated. (**c**) ^1^H-^15^N HSQC of CP12 in the presence of *C. reinhardtii* cell extract collected in the light (molecular crowding corresponding to 15 mg mL^−1^ of protein) and 20 mM H_2_O_2_ as a strong oxidizing agent. (**d**) Signal intensities for all residues in the ^1^H-^15^N HSQC spectrum of isolated CP12_ox_ (black), and in the ^1^H-^15^N HSQC spectrum of CP12_ox_ in presence of *C. reinhardtii* cell extract (green). The signal intensities are normalized against the mean of all intensities. The stars indicate residues for which the resonances are broadened beyond detection in isolated CP12_ox_ [31], and the disulfide bridges are indicated above the sequence. The CP12 sequence is indicated below the graph with same color coding as in Figure 2.

**Figure 6 biomolecules-11-00701-f006:**
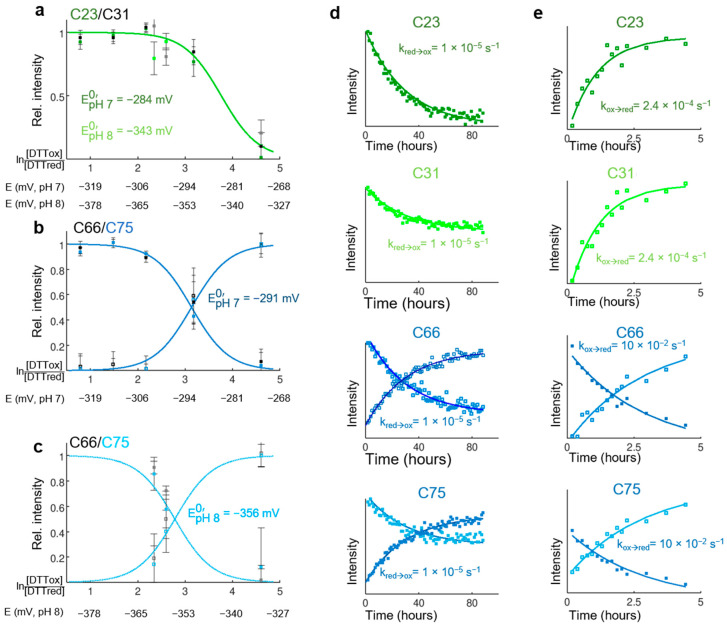
Redox transition of isolated CP12. (**a**) Thermodynamics of the redox transition for the *N*-terminal disulfide bridge. Signal intensities at NMR frequencies of reduced cysteine residues C_23_ (dark and light green) and C_31_ (grey and black), plotted as a function of the logarithm of the measured [*DTT_ox_*]/[*DTT_red_*] ratio (refer to Material and Methods). The corresponding electropotentials at pH 7 or pH 8 are indicated below. The redox titration was performed at pH 7 (dark green and black) and pH 8 (light green and grey). (**b**) Thermodynamic of the redox transition for the *N*-terminal disulfide bridge measured at pH 7. Signal intensities at NMR frequencies of reduced (filled squares) and oxidized (open squares) cysteine residues C_66_ (dark blue) and C_75_ (black), plotted as a function of the logarithm of the measured [*DTT_ox_*]/[*DTT_red_*] ratio (refer to Material and Methods). The corresponding electropotential at pH 7 is indicated below. (**c**) Same as B at pH 8. The color coding is light blue for C_66_ and grey for C_75_. The redox mid-potential of the C_66_–C_75_ disulfide bridges at pH 8 is more different than that at pH 7 than the expected −59.1 mV difference per pH unit. (**d**) Kinetic of the oxidation. Signal intensity as a function of time after buffer exchange in air-oxidized at NMR frequency of reduced C_23_ (top), reduced C_31_ (below), reduced and oxidized C_66_ (below, open, and filled square, respectively), and reduced and oxidized C_75_ (bottom, open, and filled square, respectively). The fits using the equations II0=e−kred→oxt+c and II0=c−e−kred→oxt are shown, where c is a constant, k is the rate constant of the oxidation of cysteine residues. The kinetic of the oxidation for all CP12 residues is shown in Appendix A. (**e**) Kinetic of the reduction. The color coding is the same as above. The fits using the equations II0=e−kox→red t+c and II0=c−e−kox→red t are shown, where c is a constant, k is the rate constant of the transition from oxidized to reduced cysteine residues. The reduction rate for all CP12 residues is shown in Appendix A.

**Figure 7 biomolecules-11-00701-f007:**
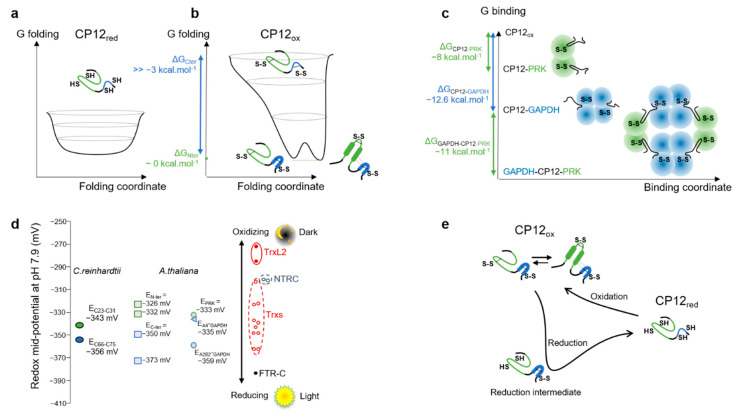
Overview of the thermodynamic equilibria of CP12. (**a**) Gibbs free energy of folding for CP12_red_. (**b**) Simplified Gibbs free energy of folding for CP12_ox_, refer to Figure 3e,f. (**c**) Gibbs free energy of binding for CP12_ox_. (**d**) The midpoint redox potentials for the CP12 *C*-terminal disulfide bridge (blue circle) and *N*-terminal disulfide bridge (green circle) measured in this study are compared to the midpoint potential for *A. thaliana* CP12 isoforms (blue and green squares), as well as for A_2_B_2_-GAPDH, A_4_-GAPDH, and PRK activity as determined in [63,64,65] (blue open triangle, blue open circle, and green open circle, respectively). For comparison, the redox potential of the ferredoxin-Trx reductase *C*-subunit (FTR-C) is shown in black, that of the NADPH-Trx reductase C (NTRC) in open blue circles, various Trx isoforms in red, as well as that of the Trx like 2 (TrxL2), as in Yoshida et al. [66]. (**e**) Kinetics of the oxidation (right to left) and reduction (left to right) highlighting the synchrony of the first, and the existence of a reduction intermediate for the second.

## Data Availability

Not applicable.

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
