# Peer review of "Flexibility of Oxidized and Reduced States of the Chloroplast Regulatory Protein CP12 in Isolation and in Cell Extracts"

_biomolecules, 2021, doi:10.3390/biom11050701_

Round 1
Reviewer 1 Report
This is a lovely piece of work, and I only have minor suggestions or points that could be clarified:
- p.1, final line: this sentence is unclear, and could be rephrased
- p.2, 'under dark...supramolecular complex': again, this is slightly unclear and could be rephrased
- Section 2.1: were integrals determined, or intensities? if integrals, were line widths fixed across different planes in the 2D as done elsewhere using nmrPipe autoFit (I don't think Sparky normally does this)?
- p.4 (and elsewhere, e.g. section 2.10): 'Delta Gibbs free energy' is a little odd, and I'd suggest should either be written as '∆G' or 'change in Gibbs free energy'
- Section 2.1, 2.2: Was the temperature 283 K or 293 K?
- Section 2.3: What temperature were experiments performed at?
- Eq. 6: What is the origin of the numerical factor 0.2? Some text and a copy of the equation are repeated accidentally immediately following the equation.
- p.7: 'Assuming that the rate limiting step for this exchange is the unfolding of the stable helical turn' - this would result in the EX1 limit rather than the observed EX2, in which unfolding and refolding occurs relatively rapidly and proton exchange is the rate limiting step.
- Fig. 2: It would be helpful to include sequence numbering in the plots, and perhaps also to indicate the formation of Cys cross-links on the sequence (where formed).
- Fig. 2b,d: I may have missed this, but perhaps some comment could be made regarding missing residues
- p. 18, end of discussion: a paragraph of instructions has accidentally been included in the text.
Author Response
This is a lovely piece of work, and I only have minor suggestions or points that could be clarified:
We thank the reviewer for his/her thorough evaluation of the manuscript; and we have made all the modifications to the manuscript he/she has suggested.
-
p.1, final line: this sentence is unclear, and could be rephrased
We have modified to: “These reversible thiol/disulfide bridges transitions imply that cysteine residues not only promote folding as they were long considered, but are also key regulatory residues. Many examples proteins containing IDRs....”
-
p.2, 'under dark...supramolecular complex': again, this is slightly unclear and could be rephrased
We have completed the sentence with the following: “Formation of these intramolecular disulfide bridges is associated with a large increase in affinity of CP12 for two enzymes of the CBB cycle, and results in the formation of a supramolecular complex.”
-
Section 2.1: were integrals determined, or intensities? if integrals, were line widths fixed across different planes in the 2D as done elsewhere using nmrPipe autoFit (I don't think Sparky normally does this)?
The referee is right, we have used the intensities. We have corrected this section.
-
p.4 (and elsewhere, e.g. section 2.10): 'Delta Gibbs free energy' is a little odd, and I'd suggest should either be written as '∆G' or 'change in Gibbs free energy'
We have made the relevant changes.
-
Section 2.1, 2.2: Was the temperature 283 K or 293 K?
The referee is right, the temperature was 283 K, we have corrected in Section 2.2.
-
Section 2.3: What temperature were experiments performed at?
The exchange was performed on ice, we have added this information.
-
Eq. 6: What is the origin of the numerical factor 0.2? Some text and a copy of the equation are repeated accidentally immediately following the equation.
The factor 0.2 is the steepness of the Nernst curve, which depends on the number of electrons and the temperature. It is actually a parameter that we have set to 0.2 to fit our experimental data, and we imposed this value for both disulfide bridges. We completed this section accordingly.
As for the repeated equation; it is actually two distinct equations. Equation (6) is for the intensities at frequencies of the oxidised cysteine residues, Equation (7) is for the intensities at frequencies of the reduced cysteine residues.
-
p.7: 'Assuming that the rate limiting step for this exchange is the unfolding of the stable helical turn' - this would result in the EX1 limit rather than the observed EX2, in which unfolding and refolding occurs relatively rapidly and proton exchange is the rate limiting step.
We apologize for this confusion, and have rephrased to: “Assuming that the rate limiting step for this exchange is the proton exchange (Figure S1)”. In figure S1, we have added the following sentence: “For CP12ox, all rates at pH 6 are significantly lower than those at pH 7, indicating that the exchange falls in the EX2 regime. This allows the determination of ΔG of folding.”
-
Fig. 2: It would be helpful to include sequence numbering in the plots, and perhaps also to indicate the formation of Cys cross-links on the sequence (where formed).
We have modified the figure accordingly, as well as figures 4 and 5.
-
Fig. 2b,d: I may have missed this, but perhaps some comment could be made regarding missing residues
We have added information about the missing residues in the figure legend: “In (b) and (g), the stars indicate residues for which the resonances are broadened beyond detection in CP12ox [31]”. We have also added this in Figure 5.
-
p. 18, end of discussion: a paragraph of instructions has accidentally been included in the text.
We apologize and we have removed it in the new version of the manuscript.
Reviewer 2 Report
The quality of experimental data and its analyses demonstrated in this manuscript is very high quality.
Although I think this manuscript is suitable for publication from journal Biomolecules MDPI, I found several issues in this as below:
- In the title and main text, "in-situ" is inappropriate. I think this study was completely "in vitro" work because the sample were based on simple "lysate". I think the authors should delete the word "in-situ" from the manuscript.
- Authors described that the HSQC experiment is "fast" and "real-time". However, the number of scan "64" is not fast and real-time experiments. The authors should clearly describe "measurement time" (how long time was take to collect one HSQC spectrum) of the HSQC in the main text.
- There are too many inappropriate manner of writing. For example; Unify representation about Sparky either "SPARKY" or "Sparky". Delete "f" between "Trx" and "and" at the line 16 on the page 2. Insert a space between "30" and "s" at the line 21 on the page 2. Insert a space between "50" and "mM" at the Materials and Methods section. Such "space issues" were found in this manuscript too many times... I cannot point out all of the inappropriate manner about "insertion/deletion of space"... I strongly recommend to the authors to do proofreading and re-editing this manuscript by order it to some professional English native speakers or company
Author Response
The quality of experimental data and its analyses demonstrated in this manuscript is very high quality.
We thank the Rewiever for these comments and have followed his/her advices for the points he/she raised.
Although I think this manuscript is suitable for publication from journal Biomolecules MDPI, I found several issues in this as below:
-
In the title and main text, "in-situ" is inappropriate. I think this study was completely "in vitro" work because the sample were based on simple "lysate". I think the authors should delete the word "in-situ" from the manuscript.
We have removed the word in-situ from our text according to her/his suggestion. We would like to note that we had defined “in-situ” in our introduction as : “with all the possible interacting partners, PTM mediators, or solubilizing molecules present in the cell extract”, and in our discussion as : “that is in CP12 physiological environment with active cellular redox-mediators”.
-
Authors described that the HSQC experiment is "fast" and "real-time". However, the number of scan "64" is not fast and real-time experiments. The authors should clearly describe "measurement time" (how long time was take to collect one HSQC spectrum) of the HSQC in the main text.
We agree with the Reviewer that there was a misleading understanding about the time of the experiments. The name of the pulse sequence is fast-HSQC, or FHSQC, as proposed by Mori et al, J. Magn. Res. B, 1995 (DOI: 10.1006/jmrb.1995.1109). The experiment is not a BEST-type of experiment. The experiment took 20 min to record. We have modified the methods section to remove this confusion.
The formation/ dissociation of disulfide bridges is much slower than 20 min, so we could measure it “in real time” even with this type of experiment. To avoid confusion, we had replaced the word real time by kinetic.
-
There are too many inappropriate manner of writing. For example; Unify representation about Sparky either "SPARKY" or "Sparky". Delete "f" between "Trx" and "and" at the line 16 on the page 2. Insert a space between "30" and "s" at the line 21 on the page 2. Insert a space between "50" and "mM" at the Materials and Methods section. Such "space issues" were found in this manuscript too many times... I cannot point out all of the inappropriate manner about "insertion/deletion of space"... I strongly recommend to the authors to do proofreading and re-editing this manuscript by order it to some professional English native speakers or company
We thank the Reviewer for his scrutiny; we have improved our writing, removed repetition of spaces, and asked a native English speaker, Pr Stephen Maberly to read our manuscript. The “f” between “Trx” and “and” is the name of the active Trx. It is a type of thioredoxins as among thioredoxins, there are f, m, y, o, h..etc….
Reviewer 3 Report
The following statements were found in the current manuscript, and those seem some leftovers. I have found a possible answer by reading the current manuscript. Please confirm it and remove it accordingly."Authors should discuss the results and how they can be interpreted in the perspective of previous studies and of the working hypotheses. The findings and their implications should be discussed in the broadest context possible. Future research directions may also be highlighted."
Author Response
The following statements were found in the current manuscript, and those seem some leftovers. I have found a possible answer by reading the current manuscript. Please confirm it and remove it accordingly.
"Authors should discuss the results and how they can be interpreted in the perspective of previous studies and of the working hypotheses. The findings and their implications should be discussed in the broadest context possible. Future research directions may also be highlighted."
We have removed this section, which was left from the guidelines to the authors. We apologize and thank the Reviewer for this.